# Racializing the Good Muslim: Muslim White Adjacency and Black Muslim Activism in South Africa

**Rhea Rahman**

Department of Anthropology, Brooklyn College, City University of New York, Brooklyn, NY 11210, USA; rhea.rahman@brooklyn.cuny.edu

**Abstract:** Founded in Birmingham, England in 1984, Islamic Relief is today the world's largest and most-recognized Western-based Islamically-inspired non-governmental organization. Framed by an analysis of processes of racialization, I argue that Islamic Relief operationalizes not a singular, but *multiple* Muslim humanitarianisms. I examine what I suggest are competing racial projects of distinct *humanitarianisms* with regards to HIV and AIDS, health, and wellness. I consider the racial implications of British state-based soft-power interventions that seek to de-radicalize Muslims towards appropriately 'moderate' perspectives on gender and sexuality. In South Africa, I argue that Black Muslim staff embrace grassroots efforts aimed towards addressing the material and social conditions of their community, with a focus on economic self-determination and self-sufficiency. I claim that the orientation of these Black Muslim grassroots initiatives denotes a humanitarianism of another kind that challenges the material and ethical implications of a humanitarianism framed within a logic of global white supremacy, and that is conditioned by racial capitalism.

**Keywords:** Islamic humanitarianism; Islamophobia; anti-Blackness; white adjacency; HIV and AIDS

## 1. Introduction

Founded in Birmingham, England in 1984 by Egyptian-born Dr. Hany El-Banna, Islamic Relief is today the world's largest and most-recognized Western-based Islamic NGO. Beginning with a humble 20 pence donation from El-Banna's young nephew, today the organization works in over forty countries and has an operational budget of over £131 million ([Islamic Relief Worldwide 2019](), p. 68). One of the ways Islamic Relief distinguishes itself among a proliferation of secular, Christian, Jewish, Hindu and other international aid agencies is through an assumed 'cultural proximity' to its beneficiary populations. However, studies of Islamic humanitarianism have pointed out that the claim of cultural proximity also assumes a homogenous Muslim *umma* that is not found in practice ([Palmer 2011]();[Benthall 2003]()). In this article, I explore the multiple humanitarianisms enacted by Islamic Relief through the lens of racialization. I consider processes of racialization *of* and *amongst* Muslims; specifically, I examine the intersections and overlaps of anti-Muslim racism, as conditioned by the Global War on Terror, and anti-Blackness, as conditioned by global racial capitalism.

Based on over two years of ethnographic research,[1] I examine organizational practices of Islamic Relief's work on HIV and AIDS to illustrate that differing responses to the epidemic are best understood as motivated by distinct racial projects. Counterpoising the institutional conditions of possibility and constraint surrounding the production and eventual removal of an Islamic Relief official policy on HIV and AIDS, with a community-oriented grassroots approach advocated by some of the organization's employees in Johannesburg, I consider competing racial projects of Muslim white adjacency versus Black Muslim liber-

---

[1] I visited offices and projects in Chad, Niger, the Netherlands, Mali, the U.K., and the U.S., and spent one full year (2012–2013) conducting research in Johannesburg, South Africa.

ation.[2] I claim that anti-Muslim racism[3] in the U.K. conditions 'soft-power'[4] ethical and political constraints that inhibit the kinds of political, ethical, and Islamic orientations the organization can adopt at an institutional, 'public' level. I argue that global anti-Muslim racism creates conditions and compulsions for Muslims around the world (but particularly those in the Global North) to align with hegemonic global whiteness (Hesse 2007; Winant 2008; Grosfoguel 2013; Grosfoguel 2016). I distinguish this white adjacency from the grassroots practices that I argue counter the implicit anti-Muslim and anti-Black racism of Western hegemonic 'development' and humanitarianism (Goudge 2003; Kothari 2006; Loftsdóttir 2009; Wilson 2012; Pierre 2020).

Central to this analysis of Islamic Relief is an understanding of race as articulated through processes of racialization, and racial projects as the building blocks of racialization processes. Racial formation recognizes race as a process that is "always historically situated, and of racial categories and meanings as fluid, unstable, decentered and constantly transformed by changing historical, social, and political relationships" (Pierre 2013, p. 4). Race takes on its varied meanings within specific historical, cultural, and political contexts. Notably, the understanding of racialization put forth in this article moves away from a conception of race as restricted to the body. While Omi and Winant (2014, p. viii) foreground phenotypical distinctions of human bodies in the social construction of race, the racialization of Muslims is more often based on what are identified as 'cultural traits' such as language, clothing and religious practices (Garner and Selod 2015). Here, I contextualize processes of racialization as conditioned by the particular logics of white supremacy (Smith 2012). Understanding racism as produced by particular logics (and not by certain categories of people or specific kinds of bodies) allows us to see the ways differently raced bodies assume differently racialized subject positions. A person racialized as a Black or brown Muslim, while subject to anti-Muslim racism, can simultaneously assume the position of the white-adjacent moderate Muslim, just as a brown Muslim can align with the religious and political commitments of Black Muslim liberation, while simultaneously benefiting from anti-Blackness. The term white adjacent has recently gained prominence in activist circles to refer to people racialized as non-white, who, in efforts to benefit from the structural advantages associated with whiteness, align themselves and assimilate into an institutional culture of white supremacy (Jones and Okun 2001). I use the term here to refer to not only people but an organization that is, by necessity, aligned with the hegemonic global whiteness that undergirds the Western development industry (Fanon 2005; Rodney 2018, 2019). I do not suggest that Islamic Relief is unique among international Muslim aid organizations or is singularly culpable for white adjacency, but to frame its ability to thrive in the Western aid industry by affirming a 'moderate' Muslim approach.

Significantly, as Pierre notes, while processes of racialization are multiple and varied, "they are all interconnected through the broader historical reality of European empire making" (Pierre 2013, p. 5). As Barnor Hesse argues in "Racialized Modernity", "Modernity is racial" (Hesse 2007, p. 643). Hesse offers an articulation of what he calls "the formal signifier of Europeanness, as a defining logic of race in the process of colonially constituting itself and its designations of non-Europeanness, materially, discursively and extra-corporeally" (Ibid. p. 646). Historicizing processes of racialization as foundational to European empire is crucial for understanding how the logics of racialization remain

---

[2] As I will develop further below, while much of the literature that examines the intersections of Islam and Black Liberation has focused primarily on Black Muslims in the United States (Karim 2005; Jackson 2009; Curtis 2012; Abdul Khabeer 2016), I extend this examination of Blackness, liberation and Islam to Muslims of continental Africa.

[3] Motivated by the *Islamophobia is Racism* syllabus, (developed by Su'ad Abdul Khabeer, Arshad Ali, Evelyn Alsultany, Sohail Daulatzai, Lara Deeb, Carol Fadda, Zareena Grewal, Juliane Hammer, Nadine Naber, and Junaid Rana), I take up the call to distinguish Islamophobia from the more accurate descriptor anti-Muslim racism. Whereas the former assumes the problem with Muslims or Islam is a problem of individual bias, the latter prioritizes analysis of historic and systematic dominance and exploitation that produce both racial exclusion and incorporation into hierarchical racist structures (Islamophobia Is Racism n.d.).

[4] Coined by American Political Scientist Joseph Nye Jr. in the late 1980s (Nye 1990), the term 'soft power' refers to the ability of a country to persuade others to do what it wants without force or coercion.

distinct, yet overlapping, in order to maintain white supremacy. After examining how the anti-Muslim racism implicit in British state funding attempts to 'civilize' Muslim perspectives on gender and sexuality, I offer a historicized account of anti-Muslim and anti-Black racism to contextualize how these racial logics operate internally, amongst Muslims, and within Islamic Relief. I examine the intersections of Muslimness and Blackness as conditioned by the distinct logics of racialization. By historically and politically situating, and thereby interrogating, these racial logics, I attempt to denaturalize them as implicit categories of analysis.[5] While anti-Muslim racism frames racialization of Muslims more broadly (compelling the good, 'white-adjacent', and moderate Muslim), in foregrounding global racial capitalism, I examine how the logics of anti-Blackness are upheld by the white-adjacent, moderate, humanitarian Muslim.

I conclude with a consideration of what I suggest is a humanitarianism of another kind. I frame the grassroots organizing of Black South African Muslim staff in Johannesburg[6] as part of a racial project consolidating a Black Muslim liberation that is distinct from white-adjacent Muslim humanitarianism practice and politics. Whereas 'development' hegemonically conceived focuses on top-down, behavior change of the 'undeveloped', I suggest grassroots organizing focuses instead on listening and responding to the material and social needs as advocated by marginalized people themselves. It is a humanitarianism that focuses on redistributing wealth and advocates for economic and social self-determination. Ultimately, I show how Islamic Relief's acceptance and ability to flourish in the Western aid industry is predicated upon its participation in racial capitalism that is grounded in anti-Blackness (Pulido 2017; Burden-Stelly 2020), an anti-Blackness that I suggest Black Muslims in South Africa resist through a Muslim humanitarianism imagined otherwise.

## 2. An HIV Policy of White Adjacency

While in the years following 9/11 many Muslim non-governmental organizations in the U.S. and U.K. faced increased scrutiny and in many cases had funds frozen due to suspected links to terrorist activity (Strom 2009; Metcalfe-Hough et al. 2015), in 2006 the British Department for International Development (DFID) established a Partnership Program Agreement (PPA) with Islamic Relief (JICA UK Office 2006). One of the strategic objectives of the agreement was for Islamic Relief to "Contribute an *Islamic perspective* to policy and research on a range of humanitarian and development issues with specific focus on HIV/AIDS, reproductive health, debt & finance and gender justice" (Islamic Relief Worldwide 2010, p. 15 emphasis added). A year later, DFID partnered with a newly instituted consortium of U.K.-based organizations linking faith and international HIV and AIDS work. The Faith Working Group focuses on what are considered 'risky' behaviors, defined in a report as gender inequality, deviant sexuality, and lack of moral foundation, which in turn are regarded as the primary drivers of the epidemic. As such, this faith-based approach to HIV and AIDS concentrates primarily on homosexuality, the permissibility of condom use, and social stigma towards those affected by HIV and AIDS. An important aspect of such initiatives is that potential solutions are framed solely in terms of behavior or culture change. A Faith Working Group report explains: "Faith is closely connected with all aspects of HIV and AIDS—it often sets the underlying values around gender and sex that influence people's behavior, and how people respond to those living with HIV" (Taylor 2007, p. 5). Given the presumption in the Western development sector of the overlap between religion and culture in the 'developing world', this working group sought to bring about behavior change by "influence[ing] religious organizations, leaders and development partners" (Ibid. p. 5) in the promotion of a U.K.-based, global HIV and AIDS strategy.

---

[5] Writing of a United States context, Su'ad Abdul Khabeer describes the ways blackness is distinguished from Muslimness by the state: "American multiculturalism defines communities strictly by their particular and non-overlapping racial identities. Thus the state can recognize Blacks or Muslims for hierarchal inclusion, but is not designed to include those who are Black and Muslim" (Abdul Khabeer 2017).

[6] Research was conducted in the period August 2012–2013, primarily in Johannesburg, with some visits to Cape Town.

The Faith Working Group partnership falls within DFID's expanding work at the time with religious-based NGOs in promoting development solutions abroad that are based on culture change. While the Faith Working Group and DFID's increased funding of faith-based organizations also targeted influential Christian NGOs, the implications of British state-sponsored funding and influence on an Islamic NGO calls for an analytic framework that foregrounds processes of racialization. Given the political context of anti-Muslim racism within the U.K. (Asad 1990; Kundnani 2007; Manzoor-Khan 2019), I frame this partnership between DFID and Islamic Relief as aligned with British soft diplomacy strategies to promote a 'moderate' Islam. In the following section, I genealogically contextualize initiatives such as the countering violent extremism programs (Countering Extremism Project 2020) that racially profile Muslims, as part of a Western 'civilizing mission' towards Islam and Muslims.[7] Significantly, I highlight the racialized element of this moderate Muslim religio-political identity as in alignment with ideological whiteness (Mills 1994; Hesse 2007), and complicit in global anti-Blackness[8] (Pierre 2013; Wynter 2003; Walcott 2014).

In a personal interview in Johannesburg in 2013 with South African Islamic scholar Farid Esack, co-founder of one of the first Muslim HIV and AIDS advocacy groups in South Africa, he suggests that Western funding plans, such as the Faith Working Group mentioned above, make the assumption that if Muslims simply gave "women a bit more right[s], and just chill out on sexuality, and make more space for condoms . . . everything will be okay." Instead of addressing the socio-economic drivers of the epidemic, Esack suggests that the 'moderate Muslim' is promoted to counter Muslims' presumed 'backwards' positions on gender and sexuality. I situate this DFID funding scheme—in the sense that it is a proposed initiative to reformulate what it means to be a 'good Muslim'—in a context of British state-sponsored counter-terrorism initiatives such as the PREVENT strategy.

Introduced in 2003 by the New Labor government of Tony Blair (and made compulsory for the public sector in 2015), PREVENT is the 'winning hearts and minds' element of the government's counter-terrorism strategy that proposes to de-radicalize British Muslims (Kundnani 2009; Qurashi 2018). As scholars have pointed out, one of the particularities of anti-Muslim racism is how it targets what are assumed to be pious Muslim behaviors—such as praying (Dandia 2020), having a beard or abstaining from alcohol—as signs of Islamic radicalism and therefore a precursor to terrorism (Mahmood 2006; Maira 2016). Soft-power tactics of the British state link the promotion of 'good' liberal and moderate Muslims in order to support Western imperialist projects both domestically (Kundnani 2014) and abroad in the so-called Muslim world (Maira 2009). As such, the effort to shape Muslim behavior by Western powers cannot be considered apart from the promotion of racialized Western imperial interests (Morsi 2017).

The racism of initiatives such as PREVENT, which operate on the presumption that all Muslims are essentially 'radicalized' and must be 'deradicalized' through programs such as PREVENT, is explicit (Qurashi 2018). However, I argue that we must understand initiatives such as DFID's Faith Working Group as also premised on the racist presumption of the need to deradicalize Muslims. Significantly, the idea that Muslims need to deradicalize is not only promoted by non-Muslims, but is taken up by Muslims themselves.[9] I maintain that for Islamic Relief to succeed as it does as one of the world's largest and most-recognized Western-based Islamic NGOs, it must promote particular humanitarian political and ethical positions that are conditioned by global anti-Muslim racism and aligned within the ideological logics of global white supremacy and anti-Blackness (Mills 1994). Though Muslims

---

[7] Note my use of genealogical so as to complicate teleological conceptions of time and history. I do not mean to reduce or collapse anti-Muslim racism as a homogenous entity that has existed, as is, for centuries, but to think through the ways the 'Muslim' is constructed as the ontological other to the Western modern subject. I aim to think through the intersections and overlaps between historicized anti-Muslim and anti-Black racism.

[8] My use of global anti-Blackness is linked to an understanding of globalized racial capitalism. While a deeper examination of anti-Blackness and racial capitalism (Bledsoe and Wright 2019) is beyond the scope of this paper, as Walcott writes, "Engaging the epistemological formations of anti-Blackness is not and cannot be merely one among other modes of thought, because only engaging anti-Blackness as foundational limit to our collective livability makes visible the overarching racial capitalist ordering of neo-colonial peoples, indigenous people, and Blacks" (Walcott 2014, p. 102).

[9] Key to this analysis is Franz Fanon's work on the internalization of coloniality (Fanon 2005, 2008).

are themselves racialized under a global racial logic that deems them the potential terrorist by a Global War on Terror, in order to operate successfully within the conditions of global white hegemony that undergirds the international aid industry (Rodney 2019), I argue that Islamic Relief at a UK-based policy level remains complicit in a global anti-Blackness that posits an apolitical and ahistorical Africa in need of white(-adjacent) saving. In the Global North, the good moderate Muslim humanitarian refutes the Muslim terrorist, but is simultaneously complicit in both anti-Muslim racism by not challenging the paradigm in the first place (Qureshi 2020), and an anti-Black white saviorism in the Global South.

Researchers at Islamic Relief's Policy and Research Division in Birmingham, England have pointed out the link between institutional funding (specifically 'Western-based' funding) and the ability to address what would be considered sensitive topics for any non-Western or faith-based NGO (Khan and Eekelen 2008, p. 3). Considering the relationship between funding sources and NGO independence and innovation, Khan and Van Eeklen discuss the importance of Western funding to allow Islamic Relief to take on issues relating to HIV and AIDS. "During 2007, the award of institutional donor funding encouraged (possibly even obliged) Islamic Relief to formulate policy positions, initiated dialogue and start projects on several important but 'sensitive' issues, including HIV/AIDS" (Ibid. p. 4). The authors address the indispensability of DFID funds to broaden their development activities on more 'controversial' social issues. A self-assessment report by Islamic Relief on DFID funding states:

> Until our PPA [partnership with DFID], we usually refrained from making statements or implementing projects in sensitive fields of work, lest we upset either our institutional partners or segments of our Muslim supporters. The PPA funding changed that risk-averse attitude as it enabled us to invest in 'having an informed opinion.' We now freely, confidently and publicly conduct research, discuss, lecture and write about issues that are controversial in Muslim circles: gender issues, reproductive health and HIV/AIDS. In line with the PPA's second and third objective, we do so in academic circles, the wider development sector, and segments of the Muslim communities in the U.K. This is new amongst Muslim NGOs: until we uploaded our first policy stances on these matters, no other UK-based Muslim organisation had publicly available policy stances on any of these issues. The PPA focus on these areas was important as the public discourse on gender, reproductive health and HIV/AIDS did not just change the extent but the nature of our impact. Today, our work is no longer limited to our programme beneficiaries but breaks taboos, counters stereotypical perceptions and fosters innovative thinking—and thus helps other Muslim organisations to address these issues in the comfort of not having been the first. Muslim Hands, Muslim Aid and several other Muslim NGOs now report on HIV-related activities. Prior to 2007, even the word HIV did not appear on any of their websites. (Islamic Relief Worldwide 2010, p. 4)

The PPA agreement with DFID specified £130,000 to organize an international consultation on Islamic responses to HIV and AIDS. The conference took place in 2007 in Johannesburg, South Africa, where approximately two hundred participants gathered at the opulent Birchwood Hotel & Conference Center. In an effort to change hearts and minds, conference planners sought to bring together people who were not typically thought to be in conversation with each other—namely, Islamic scholars, HIV and AIDS aid workers, and people living with HIV and AIDS. Activities were planned around intimate small-scale workshops in which participants could discuss controversial issues through the examination of specific case studies. The goal was to cultivate an "Islamically-acceptable and effective approach to the global HIV and AIDS pandemic" (Islamic Relief 2008, p. 4). According to a conference leaflet, organizers anticipated that answers to questions posed in the workshops would form policy recommendations to governments, Muslims leaders, and organizations working in the field of HIV and AIDS. One of the sample questions included in the leaflet asked: "In Islam, orphans (*yateem*) and other 'vulnerable groups' (e.g., *laqit*)

are clearly defined. Children whose parents are alive but incapacitated by AIDS are not part of conventional definitions. In view of the HIV pandemic, should these definitions be reviewed and elaborated? If so, what should the new definitions look like?"(Islamic 2007). Questions such as this sought to bring the level of engagement down from moral abstractions to initiate pragmatic solutions to the practical issues that afflict HIV-affected communities. As Islamic Relief founder, Dr. Hany El-Banna states in an interview from December 2007, the consultation broached topics not often discussed at an Islamic conference: "At the meeting, there was a remarkable openness and also remarkable acceptance . . . People came out and said, 'I am a gay Muslim' and 'I lived by working as a prostitute.' Of course, we were somewhat taken aback by their candidness. Certainly, in many Muslim communities such frank language is rare and often frowned upon. However, in the conference people did listen and they did accept" (El Banna 2007). He notes that the two 'hottest' discussions were around condoms and gay and lesbian issues—stating that the latter came as a surprise, "they were not planned—and caused something of a stir" (Ibid).

In an international conference that was initiated and made possible with funding from the British government through the DFID, Islamic scholars from around the world were called on to offer both 'universally'-sanctioned *and* Islamically-appropriate responses to people and communities affected by HIV and AIDS. While the event did not mark the first time Islamic scholars organized around the topic of HIV and AIDS, Islamic Relief is, as noted in their own funding report to the DFID, a trendsetter in the field of Western-based Islamic humanitarianism (Islamic Relief Worldwide 2010, p. 4). The significance and complexity of expressing an Islamically-appropriate response to HIV and AIDS revolves around the tension between advocating religiously-appropriate moral and ethical behavior while maintaining acceptance for so-called 'universal' human rights, especially as pertaining to gender, sexuality and reproductive rights. This is not to assume an inherent tension between Islamic and universal ethics, but to consider the political consequences of their negotiation.

It is perhaps not surprising that in the aftermath of Islamic Relief's HIV conference, one of the most cited public news stories concerned Jordanian-born Canadian activist Suhail Abu al-Sameed. After hearing scholars denounce homosexuality as un-Islamic and evil, al-Sameed stood amongst the crowd and announced: "As a gay Muslim, I feel unsafe, unloved, and un-respected in this space. Were I to become HIV-positive, the first thing I would lose is my Muslim community. I couldn't come to you guys for support" (IRIN 2007). Al-Sameed continued, "I wish you did not refer to gays with the (Arabic) words '*shaz*' and '*luti*'—perverts and rapists - because we are not" (Ibid.). The next morning, after offering al-Sameed apologies and warm wishes, a group of religious scholars offered a collective statement that was later published in Islamic Relief's draft HIV policy:

> Islam does not accept homosexuality. Islam does not condone men having sex with men.

> However, the aim of this conference is to fight HIV. The gay community is one of the sources of transmission. We cannot hope to fight HIV successfully if the gay community cannot be reached. Therefore, it is crucial that gay people are able to approach social workers and other sources of support without feeling unsafe.

> We, as Islamic leaders, will try to help create an environment in which that is possible.

> We hope that this will help people from the gay community, and that this will help to reduce the HIV prevalence rates.

> In addition, we hope for repentance. (Islamic 2008b)

While there are many suppositions to unpack from this passage, of note is the notion that Islam does not 'accept' homosexuality, the reduction of homosexuality to men who have sex with men, the promotion of the idea that the gay community is "one of the sources of transmission", and the hope that those who identify as gay will renounce their homosexuality and repent. While this response affirms the need to recognize that gay

men are people too and, to the extent that they are 'sick', deserve humanitarian relief, surprisingly in a conference that sought to provide practical solutions to real problems, this practical response was not informed from Islamic Relief's own on-the-ground humanitarian work working with HIV-affected people. For example, although the first reported cases of HIV were among men who had sex with men in the United States (C.D.C. 1981), the predominant mode of transmission in southern Africa is through heterosexual sex (Ramjee and Daniels 2013, p. 1). Why then would Islamic Relief, an organization that primarily provides aid in non-industrialized contexts, focus on homosexual transmission in their official policy?

Sudanese-born Dr. Malik Badri, a psychologist by training and professor at the International Islamic University in Islamabad, is the most widely-read and cited scholar on the issue of Islam and HIV and AIDS.[10] He is also the only Islamic scholar cited in the draft HIV policy. Badri defines AIDS as Gay Related Immune Deficiency—"as it was first and rightly called"—and claims that, "it is a syndrome caused by the bizarre practices of the gay revolution" (Badri 1999, p. 255). Blaming Western modernity's 'sexual revolution,' Badri describes AIDS as a "civilizationally-caused syndrome" (Ibid.). It is significant that in the passage quoted above from Islamic Relief's draft policy, it first condemns homosexuality and then specifies that the organization does not condone men having sex with men. The conflation of HIV and AIDS with homosexuality, and of homosexuality with men who have sex with men, is prevalent in mainstream debates regarding Islam and homosexuality. The stances on gender and sexuality promoted by Badri and adopted by Islamic Relief align with what Sindre Bangstad has called an Occidentalist Islamic approach (Bangstad 2009, p. 44). In an attempt to counter the anti-Black racist accusations that the virus originates in Africa (and further that it is the result of the sexual deviant behavior of Black Africans), or in an Orientalist conflation of Arab men as effeminate and gay, an Occidentalist Islam takes a stance that is at once opposed to a supposed Western culture of immoral sexuality, yet re-inscribes Western or modern categories and distinctions of sexuality (Esack and Mahomed 2011). Islamic Relief's official policy takes on a conflation of sexual behavior with identity and adopts what Ayo A. Coly has argued is a Euro-American-African *co*-production of homophobia (Coly 2013). These homophobic stances also run counter to accepted sexual fluidity in pre-colonial Islamic societies (Esack and Mahomed 2011; Mahomed and Esack 2016), as well contemporary Queer and LGBTQI Muslim activism and scholarship found throughout the world (Hendricks 2010; Moussawi 2015; Abdou 2019; Rodriguez 2020).

In that the official HIV policy of Islamic Relief, an outcome of an international conference that was itself facilitated by a British government development funding initiative, focuses exclusively on behavior change (i.e., a focus on morality and gender equality as the primary drivers and principle place of intervention to combat the epidemic), I situate Islamic Relief's work on HIV and AIDS at a policy level, within a racial project of white adjacency. Before turning to alternative models that I suggest are based in the politics of a Black Radical Muslim tradition, I first consider a genealogical account of anti-Muslim racism and its intersections with anti-Blackness in order to contextualize the broader logic of global white supremacy (Mills 2016) that founds contemporary countering violent extremism measures.

### 3. The West, Anti-Blackness, and the Ontologically 'Other' Muslim

In the previous section, I examined how a broader context of white supremacy, as manifested in the anti-Muslim racism of a Global War on Terror, influences the kinds of Islamic stances Islamic Relief can publicly take on gender and sexuality. Here I offer a genealogical account of the logics of anti-Muslim and anti-Black racism to consider how white adjacency under racial capitalism perpetuates the anti-Blackness that sustains development and humanitarianism, hegemonically conceived. As Walter Rodney summarizes in the

---

[10]  His seminal work, *The AIDS Crisis: A Natural Product of Modernity's Sexual Revolution* (Badri 1999), now on its third edition, has been translated into Arabic, Bosnian, Russian and Swahili and has received acclaimed academic rewards in South Africa and Sudan.

preface to his classic text *How Europe Underdeveloped Africa* (Rodney 2018, p. xiii), "African development is possible only on the basis of a radical break with the international capitalist system, which has been the principle agency of underdevelopment of Africa over the last five centuries." In "The Racial Vernaculars of Development" (Pierre 2020), Jemima Pierre elucidates how the unequal material relationships and processes that structure engagement between the Global South and the Global North, are upheld by a 'racial vernacular'. Exploring the co-constitution of race and language, she links the construction of African racial inferiority to the unequal material relations of resource extraction. The analysis flips the implicit premises of 'development' on its head - Africa does not need external 'saving' but rather is in 'need' due to exploitation of global racial capitalism. Anti-blackness serves as a foundation and justification for racial capitalism as African land and Africans' labor are exploited as a resource to be extracted in the service of global capitalism (Wengraf 2018). I suggest humanitarianism, though distinct from development in specific ways, is similarly grounded in a logic of anti-Blackness. I offer the following account to consider how global logics of racialization—white adjacency, anti-Muslim racism, and anti-blackness—frame different modes and outcomes of distinct humanitarianisms within Islamic Relief. I offer this account to consider what a humanitarianism that challenges the implicit anti-Black *and* anti-Muslim racism of the hegemonic Western aid industry, might look like.

I follow a number of scholars who have argued that racism is not a transhistorical category but rather a concept that takes on global significance through the world historic events of trans-Atlantic slavery and European colonialism (Mamdani 1996, 2004; Pierre 2013; Hesse 2007; Sayyid 2013; Grosfoguel 2013; Winant 2008). Many theorize and historicize contemporary racism as originating with the religious othering of Muslims and Jews during the Spanish Inquisition and/or encounters with Indigenous peoples in the Americas beginning in the 15th century. For some, it was precisely an anxiety around 'hidden Jews and Muslims'—people who converted to Christianity but retained the 'tainted blood' associated with Judaism and Islam—that serves as the foundation for scientific racism in the 19th century. However, in a revisionist historiography of a Marxist radical tradition through a historiography of the Black Radical Tradition, Robinson argues that racial thinking originates *within* Europe (Robinson 2000, p. 2). Robinson's account provides a meaningful counter to the notion that racial thinking originates with European encounters with African and Indigenous peoples in the Americas and is based primarily on phenotypical distinction.[11] Rather, race originates as the rationalization for "the domination, exploitation, and/or extermination of non-'Europeans' (including Slavs and Jews)" (Ibid. p. 27). Robinson's argument is significant for situating the origins of race to European modes of production based on domination and exploitation.

Robinson's account also provides an alternative and significant revision of the relationship between Islam and Africa, and in particular, contemporary distinctions between Muslimness and Blackness.[12] As Talal Asad (1993, 2003), Junaid Rana (2007), and Cedric Robinson (2000), amongst others, have argued, the very idea of the West emerges from and through the figure of the Muslim. It is the idea of the Muslim enemy that consolidates a unified Christendom in the Middle Ages. In the 11th century, the Crusades established a justification for just and perpetual war and later, in the 14th and 15th centuries, as secularization motivates a shift away from an idea of Christendom, the idea of a politically unified Europe emerges for the first time. Emboldened by the capture of Constantinople by the Ottomans in 1453, as religious historian Tomaz Mastnak has argued, "Western Christians were able to draw on the existing hostility toward the Muslims to invoke a sense of unity and community" (Mastnak 2003, p. 207). It is the Muslim conquest of the Mediterranean in the 7th and 8th centuries that deprived the European economies of urban life, trade and

---

[11] Significantly, this account also counters the popularized myth that racial slavery originates with what has been called Islamic, Arab, or African slavery of Muslim Arabs enslaving Black Africans (McDougall 2002; Ware 2011).

[12] In the United States, Abdul Khabeer argues that the category "Muslim" is "racially triangulated against normative ideas of Whiteness *and* Blackness" (Abdul Khabeer 2016, p. 24). In an adept study of "the relationship between race, religion, and popular culture" (Ibid. p. 10), in what she names Muslim Cool, Abdul Khabeer illuminates the move towards Blackness in the construction of U.S.-based Muslim identity.

market systems. This leads to the degradation of European life, which in turn would make the demonization of Islam and the transfiguration of the prophet Muhammad into the anti-Christ as, "tell-tale marks on Western consciousness" (Robinson 2000, p. 67). However, significantly, in this historical context, Africa was considered the place of Islamic civilization. Whereas for over 300 years prior to the trans-Atlantic slave trade Africans were a "fearful phenomenon to Europeans because of their historical association with civilizations superior, dominant, and/or antagonistic to Western societies (*the most recent being that of Islam*)" (Ibid. p. 82 *emphasis added*), this association of Africa with civilizations and in particular with Islam is later severed as a result of the European trade of enslaved Africans. Citing Hegel, Robinson states, "The wrenching of history and historical consciousness from Black people was an absolute imperative as a cornerstone for the rationalization of a slave society", and ultimately, "The African became … the object around which a more specific, particular, and exclusive conception of humanity was molded" (Ibid. p. 81). Robinson concludes that, "In the more than 3000 years between the beginnings of the first conception of the "Ethiopian" and the appearance of the 'Negro,' the relationship between the African and European had been reversed" (Ibid. p. 82). As Robinson points out, given the association of Islam with Africa, initial European voyages to the coast of West Africa were in fact inspired by European anti-Muslim sentiment (Ibid. p. 93).

Arguments of historical 'leap frogging' aside, as Trouillot reminds us, "what history is matters less than how history works" (Trouillot 1995, p. 28). Although an association of Islam with Blackness and Africa has largely been severed within global hegemonic conceptions[13], I recount this narrative to consider how we must revise imperialist historical narratives so as to consider new possibilities for collective solidarity. To historicize the category of anti-Blackness is critical to any effort to denaturalize it. Andrea Smith (2012) has argued that in the United States, white supremacy operates through the distinct yet overlapping logics of anti-Blackness/enslavability as an anchor for capitalism, genocide as an anchor for settler colonialism, and Orientalism as an anchor for war. Whereas each logic achieves a particular form of domination in relation to the nation-state, they each overlap by pitting differently racialized non-whites against each other to effectively uphold white supremacy. Whereas the anti-Muslim racism that drives the Global War on Terror is largely thought to be associated with brown (Arab and South Asian) Muslims,[14] and despite the perceived 'safety' associated with white adjacency (i.e., the idea that Muslims in the West are 'safe' from racist violence as long as they are adequately 'moderate', liberal and white adjacent), I offer this account as a reminder of the racist othering of Muslims and Blackness as paramount to an idea of the white West. That if, as Salman Sayyid suggests, "at heart of this war against terror is … an attempt to erase the contingency of the western enterprise" (Sayyid 2013, p. 15), then we must also examine the racial logics that inform Western attempts to 'reform' Islam (Esack 2013), while also confronting the ontological othering of Blackness as foundational to the modern conception of the human (Wynter 2003; Mignolo 2015). As noted above and as will be discussed further below, Muslims must also investigate their complicity in a logic of white adjacency and anti-Blackness that undergirds an apolitical and ahistorical engagement with Africa that is endemic to international development and humanitarianism (Rodney 2018, 2019). If the othering of Muslimness *and* Blackness is crucial to uphold global white supremacy, we might properly ask what are the possibilities of collective refusal of these logics of white supremacy?

## 4. The Whiteness/White Adjacency of IR

While anti-Muslim racism prefigures the Muslim man as inherently violent, alignment with the ideological tenets of global white supremacy—what I call here white adjacency—presents an opportunity that seems to curtail that racism. Perhaps proving one's good

---

[13] Of course, this is limited to a Western hegemonic perception, or what Trouillot has referred to as a West's geography of imagination (Trouillot 2004, p. 8). Of course millions of Black African Muslims would and do deny such a severing.

[14] However this is not the case as Black Muslims are also subject to the surveillance, deportation and incarceration enacted by the Global War on Terror.

humanitarian Muslimness may present a way out from being racialized as the bad Muslim terrorist? For a more specific look at the racial politics that play into the development of an HIV and AIDS policy, I consider Islamic Relief's own raced, classed, and gendered good humanitarian, ethical positions.

During my first visit to the Islamic Relief-USA head office in Washington D.C. in 2009, in response to my question about push-back from Muslims who argue that Islamic Relief should focus solely on helping other Muslims, an upper-level administrator tells me that, "In Islamic Relief we are equally in the business of educating Muslims of proper interpretations of Islam, as we are in the business of helping those less fortunate." The question remains, on what kinds of raced, classed, and gendered basis does Islamic Relief base their own 'proper' interpretation of Islam? In Islamic Relief's early days in the 1980s, the organization's founder Dr. Hany El-Banna would arrange study circles with the local Muslim community in Birmingham, England. In them, he emphasized the importance that the separation of humanitarian work from proselytization is itself an Islamic value. In an interview sited in his biography, he states, "Under no circumstance should humanitarian work be mixed with promoting religion. Faith-based for us means translating our faith into action as our faith inspires us to help all those who are poor and vulnerable, not simply those of a particular religious denomination" (quoted in Din 2011, p. 42).

This commitment to impartiality has served Islamic Relief well in the Western aid world. However, this good standing in the West does not come without political, ethical, and religious constraints. When Islamic Relief provided aid to populations in Iraq during the first Gulf War, the organization was intensely scrutinized to ensure that they were not supporting any one faction in the war. As noted above, in the U.S. in the years following 9/11, as a number of Islamic organizations found their funds frozen due to suspected links to terrorism, Islamic Relief saw their annual donations increase three-fold (Strom 2009). Given that one of the five pillars in Islam is to give zakat, as Muslims in the U.S. saw their options for places to donate shrink, and even worried they themselves might be implicated if they donated to an organization that the U.S. deemed 'terrorist', more funds were channeled to Islamic Relief. A 'good' (moderate) interpretation of Islam is necessary not only for Islamic Relief to function in the Global North, but also so that Muslim donors can give without fear of repercussion. This is why a good reputation is not simply beneficial, but necessary for their ability to operate and function in the U.S. and U.K., and therefore globally.

In an online interview, El-Banna describes the difficulties of gaining credibility, "particularly if you call yourself Islamic in the heart of the Christian civilized western world. [We had to ask] 'How can the West accept you and the Muslim not suspect you?'" (El Banna 2016). Attaining the status of a trusted figure by Muslims around the world, and the development sector of the British government, it would seem El-Banna, and by extension Islamic Relief, succeeded in getting the West to accept and the Muslim to not suspect. But the question remains, which Muslims? Which Muslims are worthy of being 'saved' within a logic of white adjacency necessary for the West's trust, and who is excluded? For an expanded conception of the worthy beneficiary, in the following concluding section, I turn to an Islamic humanitarianism imagined otherwise by Black South African Muslims.

## 5. A Grassroots Racial Project

After having to curtail my research in Mali after a military coup in Bamako in March 2012, and before moving my field site to South Africa, I spent the interim in Amsterdam. I contacted the Islamic Relief-Netherlands fundraising office to gain another perspective on IR's fundraising partners in Europe. The program manager at the office—a serious and genial Dutch woman named Anna—invited me to the office. Before working for Islamic Relief, Anna, who was the only non-Muslim working in the office, previously worked for an NGO that was closed due to drastic national funding cuts in the Dutch development sector. Anna maintained that IR's private funding scheme was one of the advantages of religiously-inspired aid. As opposed to her previous job, Islamic Relief was less reliant

on government funds and therefore less subject to fluctuations in government-sponsored international development. However, as mentioned above with regards to DFID funding, this also meant being subject to the interests of private Muslim donors.

In addition to managing IR-Netherland's relationship with program offices abroad, Anna also managed their participation with the Religion and Development Knowledge Center (RDKC). Like the U.K.-based Faith Working Group discussed earlier, the Religion and Development Knowledge Center is a Dutch government-funded initiative dedicated to approaching international development through the lens of religion. One of the RDKC's administrators explains to me that whereas previously all the participating organizations were Christian, the RDKC was eager for a Muslim representative. Islamic Relief was the only Muslim organization to partner with the center, and in fact this participation was largely dependent on Anna's own interest. Islamic Relief's contribution to the working group was to host an annual conference at the center. Given that I was soon to leave Amsterdam to conduct research with IR-South Africa—where it seemed most of Islamic Relief's work on HIV and AIDS is located—and could act as a liaison to help Anna organize on-the-ground resources for the conference, Anna decided to focus that year's conference on Islam and HIV and AIDS.

I offered to help Anna in the organization and implementation of the conference. In order to conduct research on Islamic Relief's work on HIV and AIDS in South Africa, and to interview and select a speaker working in South Africa to speak at the conference in Amsterdam, Anna joined me in Johannesburg. She felt strongly about inviting a staff member from South Africa, not only to offer conference participants in Amsterdam the perspective of someone working with communities on the ground, but for staff and supporters of Islamic Relief in the Netherlands to have a first-hand account of how Islamic Relief is addressing the epidemic. In South Africa, Anna and I began by interviewing all staff working on issues related to HIV and AIDS. At the time, we realized that the aforementioned 'official' HIV policy that was facilitated by the DFID funding program had been removed from the Islamic Relief Worldwide website. While all the staff with whom we spoke in South Africa maintained that it would be good for Islamic Relief Worldwide to take an official policy stance on HIV and AIDS, many also seemed to take advantage of the lack of an organized and structured policy on HIV and AIDS and pursued their own initiatives.

Anna and I quickly discovered that Islamic Relief-South Africa's work on HIV and AIDS seemed to differ according to whom we spoke. This is in contrast to IR's draft HIV policy that states that while other faith-based NGOs often allow for discrepancy between their formal stance and their actual service delivery, Islamic Relief does not wish to reproduce such an inconsistency: "Wherever possible, IR will *not* take such a low-profile or indirect approach, as IR feels that its stance is as important as its actual practice because of the trickle-down effects this stance is expected to have on Muslim communities and other NGOs who look to IR for guidance on sensitive issues" (Islamic 2008b).

One of the first people we interview is Abdul. Abdul is Xhosa and reverted[15] to Islam as a teenager. He was hired by Islamic Relief in 2010 as an HIV and AIDS and gender-based violence (HIV/GBV) officer. However, under the leadership of a new country director a year later, he was shifted to a position in marketing doing 'donor care.' As an HIV/GBV officer, Abdul did HIV/AIDS training and development, and mobilized and networked with different HIV stakeholders within the NGO sector and South African government. He excitedly describes his success at establishing relationships with other faith-based organizations and government agencies. He also conducted trainings in prisons for predominantly Muslim inmates, for imams, for Muslim and non-Muslim religious leaders, and for gender-based groups, both for a women's empowerment organization and

---

[15]  Many Black Muslims in South Africa maintained that they are reverts, not converts, to Islam. Those who prefer the term revert do so based on the Muslim belief that all people are born with a natural faith in God. According to this belief, children are born with an innate sense of submission to God, which is called the *fitrah*. Some people, then, see their embrace of Islam as a return back to this original, pure faith in our Creator.

a men's gender justice organization. I recall he was one of the first aid workers I came across for whom gender did not simply mean women. He explains that he is passionate about teaching men their role in HIV prevention and the elimination of gender-based violence.

During our interview, he tells us that he now feels blocked. He was told to stop networking because "it brought no fruits." He says "So I asked, 'what type of fruits are we talking about? Is it that you want money and then we can say it does benefit? Does it mean that IR's purpose is to bring money?'" Abdul explains that social networks and connections are more important than funds:

> Let me tell you something about programs, especially HIV/AIDS, if you want to save costs, you partner with peoples. For example, we want AIDS workshops targeting religious leaders, Muslims and non-Muslims. We look at who else is dealing with the same thing. We create partnerships with them. And then we create our own plan. To say: we all want to achieve this goal. We would do capacity building, skills assessment, needs assessment. We then tap into what we have and what we can get. Because nowadays, to avoid duplication, you need to partner with people. You need to use other peoples' resources.

One of the organizations he works with is the South African branch of the International Network of Religious Leaders Living with or personally affected by HIV and AIDS (SANARELA+). In 2010, the organization created the SAVE Toolkit—a workbook designed for leaders in various faith communities to address HIV and AIDS in their respective communities. In 2012, at an HIV/AIDS and gender-based violence training-of-trainers seminar by Islamic Relief and sponsored by UNAIDS, Abdul discusses this partnership with attendees. The workshop was a three-day meeting attended by over fifty Islamic Relief staff and imams from across Africa. A representative of SANARELA+ introduces the toolkit and explains how it is different from other HIV resources. The second edition of the toolkit states:

> At its heart, SAVE is a prevention methodology that takes account of the drivers of the epidemic. It provides a space to explore the unmentionable subjects of sexual practice and embedded cultural practices that lead to new infections. It assists communities in challenging the systemic factors that lead to new infections. It assists communities in challenging the systemic factors that lead to spirals of poverty and abuse. In short, it challenges human beings to be humane. (INARELA+ 2018)

At the conference, an imam from Mali asks whether they experienced push back from Muslims where they have already promoted the toolkit. Abdul confirms that yes, there were issues, particularly surrounding the promotion of condoms (including the use of condoms within marriage) and concerns with LGBTI issues. The SANARELA+ presenter affirms that the toolkit is not promoting certain activities or any particular sexual orientation but that, "whether or not you support people of a certain orientation—we are entitled to our own opinions - but if people are being tortured and abused because of their sexual orientation, do we have a right as a religious sector to protect those people? That is the difference." Abdul continues, "when I work with Muslim communities, I have the relevant *hadith*, I have the relevant *ayas* of the Qur'an . . . this toolkit was developed to deepen these conversations. I was involved with creating this toolkit and I know the importance of it and how it has changed people's lives . . . A lot of the *ulema* took issue in South Africa when we first started working on this, but we are working tirelessly that this is a success." At the end of the session, a Somali aid worker from Kenya thanks the presenters and says, "I just want to congratulate them, because everything you do has its challenge. Nothing goes smoothly in this work but I take it as a positive. If you have challenge you are doing something, when you are not having a challenge, then you are doing nothing." Islamic Relief-South Africa partnered with SANARELA+ because of Abdul's relationship with them. While Islamic Relief was not the first Muslim organization to address HIV and

AIDS in South Africa (Ahmed 2000), Abdul's networking pushed Islamic Relief towards a more 'humane' response to HIV-affected people than was expressed in the language of the 'official' HIV policy that hopes for repentance for gay Muslims.

Abdul recalled a telling anecdote about his previous job that revealed a common theme for faith-based HIV work. The Muslim AIDS Project, an affiliation of the Jamiat Ulema of Johannesburg, the South African Indian-led Islamic 'ruling body' of Johannesburg, promoted sexual abstinence and 'being faithful' as its primary counters to the epidemic. I mention to Abdul that the stance of the organization seems out of line with the experiences of the communities he works with, and yet the way he spoke about his community activism sounded pragmatic and realistic to the actual situations in which people found themselves. He explains that when one goes to a community with the intention of changing others, one has to realize that is the others who change you. "They bring reality to you. Situations bring reality to you. Your experiences also educate you, or informs you, [but] whatever position you are taking, it is not relevant. And you also learn from other people doing. And you get to understand that whatever we are talking about here [in the office], it is different from what is happening [in the community]." He had no problem with the discrepancy between MAP's official stance promoting abstinence and faith and the honest conversations he was having in the field. When I asked whether the board at MAP would be upset that he was not promoting abstinence in the field he replied, "I would write what they want but I would do what I just said." He explains that MAP's ABC strategy—abstinence, be faithful, and only then promote condoms—could maybe work for primary students, but not secondary school students: "you can never say that in the townships. Try it! Go to a township in Soweto and say: you must abstain. I tried it. I went there, when I spoke about abstinence this student looked at me and asked me a very funny question: 'you want to deny us from what you have had already?'"

At both the Muslim AIDS Project and Islamic Relief, Abdul towed the official line, no matter how 'out of touch with reality' and then acted in response to pragmatic community needs on the ground. I read his ethical stance of listening, radical openness, and responding to the needs of his community, as related to his ethical and political orientations as a Black Muslim. In one of my first conversations with Abdul, he tells me that he is unapologetically pro-African. He says that Black South Africans suffer from an inferiority complex "they want to be like Europeans, Americans. What I'm instilling to my child—'you're Black, South African, you are not less superior to anyone. You are loved, raised your head high.' You need to give love" he tells me, "that is transformation." But Abdul gave more than immaterial 'love'. Before my arrival in South Africa, before he was moved to 'donor care', in addition to the HIV networking and trainings mentioned above, he was also running an after school program for youth in the surrounding neighborhood of the administrative office in Fordsburg, Johannesburg. He recruited student volunteers to come in on Saturday mornings to tutor younger children. He solicited local women to donate and cook food. He explained to me that the program did not cost the organization anything. But after the change in management, he was told to close down the program for lack of funds. Abdul concludes that he and his initiatives were not a priority for the organization. In frustration, he explains, "We should have a proper vision of what are the priorities, what are the focus projects and interests of IR and how do we want to achieve it within South Africa." Abdul believes that as Muslims, they are to be pro-active, "Because I believe the Prophet was pro-active. Most of the work that he did, he never spoke much, but he did much. And that is what attracted people to Islam." For Abdul, who as he mentions is also inspired by Elijah Muhammad and Malcolm X, his Islamic values pushed him to *do* and not talk. For Abdul, *doing* meant going into his communities, being open to being changed and formed by the realities faced by his community, and giving back what he could. He was involved in organizing and responding to his community, as opposed to offering charity.[16]

---

[16]  For more on Islamic Relief's approach to zakat, often translated as charity, see (Rahman 2017).

When I returned to Johannesburg in July 2019, I was unable to meet with Abdul, but had heard from others at Islamic Relief that he was still involved with Black Muslim organizing and activism in Soweto. Since my first trip in the period 2012–2013, the formation of the Gauteng Shura Muslim Council, and the first Black South African Muslim conference held in Johannesburg in April 2019 indicate a further specialization of Black South African Muslim organizing around cultural, ethical, and practical concerns as distinct from Muslims of an Indian or Cape Malay background. Many Black South Africans Muslims I speak to in 2019 describe similar issues. An imam who ran a mosque and community center in a township outside Johannesburg explains to me three sources of alienation faced by Black South African Muslims. First, from one's own family. He says that most Black South Africans come from an atheist or Christian background, so when one reverts to Islam, they are considered an outsider in their own family. Second, one is alienated from one's surrounding community. Legacies of apartheid mean that after twenty five years of democracy, most South Africans still live in racially segregated communities. When Black South Africans revert to Islam and adopt Arabic- or Indian-style clothing to signify their Muslim identity, many face discrimination from peers. He tells me in the township he is often asked "Why are you dressed like an Arab/Indian?" With increasing accounts of xenophobic violence towards 'foreigners', Black South African Muslims face accusations that in becoming Muslim, they have sided with either the Indians, Colored/Malay, or foreign African Muslims from throughout the continent, and are therefore not authentically Black South African. And third, the racism Black Muslims face from Indian and Colored/Cape Malay South African Muslims.

When I visit the new Islamic Relief office in different location Johannesburg, I meet Da'wud, a Programs Coordinator who tells me that he only experienced interpersonal racism *after* he reverted to Islam. He says that for him growing up in the township of Soweto with politically active parents, he was prepared for or rather expected a certain kind of racism from white South Africans. However, he describes the unique shock and disappointment the first time he visited a mosque and an Indian Muslim refused to shake his hand. However, he also maintains that, "on the one hand, calling Indian Muslims racists . . . it's an easy target, it's obvious. But we also have to understand the structures behind these behaviors. This Indian Muslim is acting in accordance with the beliefs and practices he grew up with." It is not that he accepts the interpersonal racism from Indian Muslims— he had recently pulled his daughter from the Indian Muslim school she was attending to protect her from the kinds of demeaning and psychologically detrimental experiences he faced from Indian Muslims—but he also says, when he faces a condescending question such as when a non-Black Muslim asks "How did you become Muslim?", he says at first his response was to counterattack. He finds the question condescending in that he feels he is being asked to prove himself a Muslim, instead of being accepted unquestioningly. But he says he now understands that to counterattack is pointless. He talks instead about transformation. He explains that South Africa has never dealt with racism at a national level. When I ask about the Truth and Reconciliation Commission, he brushes it off as a farce, in which perpetrators were left 'scott-free.' But he says change must be strategic and transformative.

Like the aforementioned programs initiated by Abdul before him, Da'wud also started a grassroots empowerment program. At the community center called Osizweni—a Zulu word meaning 'Place of Help'—that Islamic Relief sponsored in the township of Ennerdale outside of Johannesburg, Da'wud created and ran a Women Empower Project. Wanting to create a more self-sustaining financial model for the guardians of sponsored orphans, the program targets approximately 120 woman, 60 percent migrant and 40 percent local. The township of Ennerdale was established in 1950 as a Colored Area in the Groups Areas Act introduced by the apartheid government, but has since seen an influx of domestic migrants from the Eastern Cape, as well as international migrants from Lesotho and Zimbabwe. Racial tensions between Colored and Black South Africans, as well as xenophobic violence, are some of the many social conditions to affect Ennerdale. Da'wud, as well as the Zulu

woman who initiated and directed Osizweni, sought to identify and support the most marginalized in their community and identified migrant women as having the most difficulty as they did not have access to government support and were more often subject to structural and interpersonal violence. The program offered skills development such as gardening and sewing, political education on workers' rights, business management and microfinance, and social programs focusing on social cohesion to aid tensions between various marginalized communities.

In these efforts, Da'wud talks a lot about self-determination and self-sufficiency. He and I bond over our frustrations and critiques of models of charity that sustain unequal power relations (Rahman 2017). Da'wud and others complain to me of the problematic racial optics and domination of wealthy Indian South African Muslims who build mosques, madrasas and community centers and come to deliver food hampers and school supplies in predominantly Black townships. People express frustration that when Indian South Africans bring their money into Black townships, they also bring their own distinct culturalist interpretations of Islam. Another Black Muslim employee at Islamic Relief laments that when Indians come to speak about Islam in Black townships, they do not understand or respect their African culture. She explains, for example, they criticize the concept of dowery without an understanding the importance of *lobola*[17]—a concept of significance in Zulu, Xhosa, and Shona or Ndebele traditions. However, Da'wud is also astute to the legacies of apartheid that influence social divisions over solidarity and cohesion, i.e., the divisions between racial categories but also within them. He notes significant tensions between various sects of Black South African Muslims. While hopeful about increased Black Muslim organizing, he is weary of increased divisions. "We are collectively Muslims, not at the expense of the other. I don't have to insult another to get recognized." He is less concerned with interpersonal rifts and more invested in economic justice. Returning to the account of the Indian man who refused to shake his hand, he also explains the problem is more about the materialist structural conditions of Black workers subject to Indian business owners. His efforts within Islamic Relief were about addressing these economic standards and shifting the organization's programing from charity to economic self-sufficiency.

While Da'wud was not as actively engaged in HIV and AIDS programing as Abdul, I include his activism alongside an account of Abdul's as they both represent attempts at restructuring and reimaging a Muslim humanitarianism otherwise—a humane humanitarianism that is not 'top-down' but that foregrounds a grassroots approach that centers community self-determination. As with Abdul, Da'wud is concerned with listening to his community and providing strategic interventions that lead to economic self-sufficiency, which in turn he sees as pivotal to community and religious self-determination.

## 6. Conclusions

To conclude, I return to my initial visit to South Africa when, despite her efforts, Anna was unable to gain permission from Islamic Relief-South Africa to bring Abdul to Amsterdam to present at the HIV and Islam conference. Although she wanted someone working on the ground in South Africa, in the end she was able to secure Islamic Relief's founder Dr. Hany El-Banna as keynote speaker. At the conference in Amsterdam, after El-Banna presents on the importance and challenges of working on HIV and AIDS as an Islamically-inspired NGO, during the Q&A, many questions from the largely non-Muslim audience revolved around how to change the Muslim communities' perceived slower and problematically regressive response to the pandemic (as opposed, apparently, to other faith communities). One attendee pressed on how exactly to bring about changes in the Muslim community. El-Banna affirms that change in the community must come through engagement with key players in the community, even if those key players are offensive to universalized Western liberal sensibilities. "That is why I said: are we afraid to sit down with Taliban? Because we might not get funding from the United Nations or European

---

[17] For an African feminist take on lobola, see (Msimang 2002).

Union anymore. But this kind of marginalization does not bring us to the solution." Here he reflects precisely on the institutional constraints of global white supremacy. He recognizes that, of course, if Islamic Relief were to engage with so-called 'bad Muslims', they would lose their support and prominent standing in the Western aid world. In other words, they would fail in their efforts of having the 'West accept, and the Muslim not suspect'. However, El-Banna's suggestion of engaging with the Taliban, for example, is not suggested as in the radically open approach enacted by Abdul and Da'wud, but more of a top-down, influencing and educating 'bad Muslims' of the 'proper' interpretation of Islam. El-Banna says that the Muslim communities that need changing are the 'isolated communities', suggesting that, "they need exposure to other possibilities. It's providing them with a wider perspective, so that they themselves can open their perspective, to break out and see that there are other solutions. Ownership is important, but part of the role of outsiders is to bring in people that can widen the perspective and open the view." Whereas this article has examined how behavior change suggested by Western liberal actors want to 'civilize' HIV and AIDS out of the Muslim community, I suggest that in its white adjacency at a policy level, Islamic Relief takes on the role of civilizing from within.

In a personal interview, Farid Esack offers a different perspective on the question of insider and outsider influences in relation to Muslims, Islam, and development.

> You see, I must make a distinction between the whole idea that Muslims must be pushed into change. That I support, but that changing must be always in response to internal damage, internal inconsistencies and internal pain. That push must never come from external impulses that wants to civilize these Muslims . . . I think changes inside the Muslim community must be driven by its own internal imperatives of pain, and injury, to its own marginalized communities.

With regards to El-Banna's suggestion to engage 'isolated Muslim communities' to be more open, it is imperative to consider where and how this change is politically, ethically, and racially motivated. Esack connects the problematic *external* attempt to change communities to what he sees as the broader problem of international development.

> But this also brings into the front the whole problem of development. When development is seen of a project out there, rather than a transformation of everything, including a quality of life in the North, in order that the quality of life in the South will improve. So all notions of development intersect with notions of otherness and purity and so on. And all of this means that you can't actually produce a [official policy] document. You can move towards a document, but once you are going to produce a document that is going to see the day of light, these tensions are going to be too strong. And the organization is too fragile. Given the fact of HIV work on the ground in South Africa and all the other elements in the organization, bureaucrats and the people that must go out there for money, the big donors and so on. When you've got the document in front of you, then all those tensions that you could negotiate quietly, work around, and in South Africa you don't have to put too specifically into your report that you've been distributing condoms and so on, a written report is very difficult, because it lays to bare all of these tensions.

On the one hand, it is perhaps obvious that, at an institutional level, Islamic Relief must make certain concessions to uphold its reputation and ability to function within the Western aid industry. Islamic Relief must enact the good, moderate, liberal, white-adjacent, Muslim humanitarian. Perhaps it is also obvious that on the ground in the places that Islamic Relief delivers aid, there are quiet negotiations that engage controversies that, institutionally, IR cannot maintain, such as handing out condoms and supporting LGBTI Muslims. But given the genealogical, historicized account of the othering of Muslimness and Blackness offered here, and the consequences in terms of the enduring violence of coloniality and racial capitalism that maintains a system of devastating inequality, we might ask what 'good' is achieved when IR upholds a logic of white adjacency? We might

see these competing racial projects of white adjacency and Black liberation as symbiotic to the extent that Abdul would write in reports whatever he knew his bosses wanted to hear, and on the ground responded to the actual needs of his community. Conceivably, at best, Islamic Relief's white adjacency could be exploited to secure Western aid that can then be redistributed to the Global South in ways that disrupt global hegemonic white supremacy. Yet we must also ask of the ways that white adjacency maintains complicity in the very anti-Black system of global racial capitalism that produces the inequality that 'development' and 'humanitarianism' supposedly seek to dismantle.

I conclude with these varying accounts of insiders and outsiders with regards to development and change amongst Muslim peoples to reaffirm a commitment to analyze the historicized, political contexts within which insiders versus outsiders are determined, and the question of one position's relative influence over another. My work recognizes the logics of racialization as foundational to constructions of insider and outsider. I began my research with Islamic Relief with the question of what it means to 'do good' and conclude that we must consider 'the good' in particular historicized relations of power. A genealogical account of Muslimness and Blackness is crucial to foreground coloniality, racial capitalism and anti-Blackness as the context within which battles over the 'good' take place. Reading Islamic Relief's institutional success in the Global North as in alignment with the ideological position of global white supremacy is certainly not an indictment of the people working for and with Islamic Relief who genuinely strive to improve the conditions of people around the globe, and without whose time, consideration and support, this research would have been impossible. Rather, a focus on the intersecting logics of racialization, and an ensuing examination of the structural and institutionalized conditions of possibility within the global aid industry itself, reveals limitations of what 'doing good' can look like. Privileging global white supremacy, anti-Muslim racism, and anti-Blackness in this analysis allows a vision that does not collapse difference, but following Da'wud's indictment to focus on transformation and emphasizing the grassroots approach of Black Muslims in South Africa, seeks out the intersections of exploitation and exclusions to affirm solidarity against forms of oppression that urgently threaten the livability of our planet, and the multitudes of people striving to survive within it.

**Funding:** This research was funded by Wenner Gren Foundation for Anthropological Research, The New School for Social Research Dissertation Fellowship, and the PSC-CUNY Award # 62694-00 50.

**Institutional Review Board Statement:** The study was conducted according to the guidelines of the Declaration of Helsinki, and approved by the Institutional Review Board of The New School for Social Research (Approval ##111-2011, 6 October 2011). An exemption was granted by the Institutional Review Board of The City University of New York on 12 June, 2019 due to the following criteria: (i) The information obtained is recorded by the investigator in such a manner that the identity of the human subjects cannot readily be ascertained, directly or through identifiers linked to the subjects; (ii) Any disclosure of the human subjects' responses outside the research would not reasonably place the subjects at risk of criminal or civil liability or be damaging to the subjects' financial standing, employability, educational advancement, or reputation.

**Informed Consent Statement:** Informed consent was obtained from all subjects involved in the study.

**Data Availability Statement:** The data presented in this study are not publicly available due to confidentiality.

**Acknowledgments:** This research would not have been possible without the time and support of Islamic Relief staff and volunteers in the United States, the United Kingdom, the Netherlands, and South Africa. I would also like to thank Professor Farid Esack and the Decolonizing Religion reading group at the University of Johannesburg. Research for this article was funded by the Wenner Gren Foundation for Anthropological Research, The New School for Social Research Dissertation Fellowship, and the Professional Staff Congress Research Award. This article has benefited from the thoughtful readings and comments of Marisa Solomon, Lawrence Johnson, and Naomi Schiller. I am also grateful to the participants of the Africa, Globalization and Muslim Worlds Conference at the

Harvard Divinity School, 19–21 September 2019, where this work was initially presented. The Critical Muslim Studies Summer School in Cape Town in January 2020 also contributed to advancement of this article. I would especially like to thank the anonymous reviewers at *Religions* whose thoughtful suggestions contributed to the organization of this chapter.

**Conflicts of Interest:** The author declares no conflict of interest.

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
