# Peer review of "Racializing the Good Muslim: Muslim White Adjacency and Black Muslim Activism in South Africa"

_religions, doi:10.3390/rel12010058_

Round 1
Reviewer 1 Report
This paper begins very promisingly with a deft framing of the politics of Islamic humanitarianism in terms of “competing racial projects of Muslim white adjacency versus Black Muslim Liberation” (line 39). The author makes a persuasive case that “thinking of Blackness and Muslimness as distinct identities obscures possibilities for recognitions of overlapping systems of exploitation and solidarity in struggle” (lines 311-312). As the more ethnographic sections of the paper unfolded, though, I found myself frustrated by a lack of specificity about what “Black Muslim Liberation” does or should look like in the context of HIV/AIDS prevention efforts in South Africa.
In particular, after the theoretical buildup, it was a letdown to read the description of Zahir’s “grassroots empowerment program,” which consists entirely of the statement that it is “based on an antiracist do-for-self strategy that refutes the subtle (but no less insidious) forms of white supremacy as that of British-government funded [programs?] that comes with its ‘making moderate Muslims’ stipulations” (lines 534-536). I need to know much more about what Zahir is doing, and how it differs from what Abdul does in his capacity as an Islamic Relief community organizer who sends palatable reports to headquarters while “using his charm and enthusiasm to mentor young Black men” (lines 494-495). In particular, what makes Zahir’s approach to issues of sexuality and disease prevention “antiracist”?
More broadly, the author needs to explain whether (and if so, how) tendencies to think of “Blackness and Muslimness as distinct identities” inform Islamic Relief’s HIV/AIDS prevention work in South Africa. Given the author’s theoretical intent to bridge this gap, this seems a significant ethnographic lacuna. (Incidentally, he or she should not use the term “Black Africa” [line 340], a notorious conflation of geography and phenotype, especially while decrying “white-adjacency.”)
Another point of analytical slippage or lack of clarity occurs in the conclusion, where the author refers to Farid Esack’s discussion of “internal” versus “external” attempts to “change [Muslim] communities.” I am not sure whether the author endorses this distinction or means to critique it. If the former, who is to say what counts as internal as opposed to external? If the latter, what is the ground of criticism?
In conclusion, this paper makes some interesting theoretical gestures but falls short in ethnographic follow-through.
Reviewer 2 Report
I believe this article is quite well-done, and look forward to it's publication. I will likely use it in courses on ethics or Intro to Islam in looking at societal issues.
I felt the organization flow worked well, too.
I noticed a few formatting issues (sometimes there aren't paragraph indents, or sometimes there's double or single quotation marks, or there are a couple paragraphs where the first line is disconnected from the body), but I assume these will get worked out in editing.
Great work!!
Author Response
I have addressed formatting issues such as paragraph indents and double and single quotation marks, for consistency.
I am grateful for Reviewer 2's comments and consideration of my work.
Round 2
Reviewer 1 Report
This paper shows substantial improvements from the previous draft. In particular, I am happy to see that the author has added substantial concrete material concerning the grassroots community work that Zahir and Abdul are carrying out in South Africa. The concluding section on discourses of “internal” versus “external” sources of change is clearer as well.
My remaining difficulties turn on matters of interpretation. The author casts Zahir and Abdul’s efforts in entirely oppositional terms vis-à-vis the Muslim humanitarian policy positions of Islamic Relief and comparable organizations: “I show how Islamic Relief’s acceptance and ability to flourish in the British aid industry is complicit in global anti-blackness that Black Muslims in South Africa … defy through a practice of Muslim humanitarianism imagined otherwise” (lines 67-70, my emphasis). Is “defiance” really the most appropriate assessment, though? Later on, the author points out that “At both the Muslim AIDS Project and Islamic Relief, Abdul towed [should be “toed”] the official line, no matter how ‘out of touch with reality’ and then acted in response to pragmatic community needs on the ground” (lines 528-530). Clearly, this is a form of accommodation rather than defiance. I would suggest that accommodation likely operates in the other direction as well, as IR officials say what their donors and sponsoring government agencies want to hear while turning a blind eye to the ways Zahir and Abdul engage LGBTI communities. The author tells us as much in the Conclusion (lines 695-698). A more generous reading of el-Banna’s approach would stress that he has to navigate carefully among numerous institutional and political imperatives. Does the fact that IR is forced to navigate in these ways really give rise to strict “limitations [on] what ‘doing good’ can look like” (line 723)? The fact that Zahir and Abdul are able to carry out their work as they see fit would seem to suggest otherwise.
The author has responded to my suggestion to elaborate on his or her original concern about “Blackness and Muslimness as distinct identities” by removing this phrasing and introducing a summary of Cedric Robinson’s reading of the history of racial imaginaries in Section 3. I’m afraid I regard this decision as a poor one. The intersections between Blackness and Islam do seem relevant given the discrimination that African converts to Islam encounter at the hands of other Muslims in South Africa, and it would be a good idea to say more if possible on a general level about how IR and/or community workers address this issue. Whether or not the author wishes not to pursue this particular avenue of inquiry, I would advise omitting the two paragraphs on Robinson. Apart from the fact that other historical readings have been advanced, I’m not persuaded that such a broad genealogical account of anti-Muslim racism bears directly enough on the particular predicaments of accommodation faced by humanitarian workers. If the author decides not to consider at a general level how IR deals with intersections between Blackness and Islam in South Africa, I would advise omitting Section 3 altogether.
Please be on the lookout for misplaced modifiers such as “Operating in over 40 countries, I argue that …” (line 7).
Note that the term “tenants” should be “tenets” at line 359.
Please remind the reader at lines 432-3 what the relevant “official HIV policy” had been.
Author Response
I have attached my point-by-point response to the reviewer's comments together in the same document as my revised manuscript. The response to the reviewer's comments are included at the end of the document (pages 24-26).
